# Improving Blind Spot Denoising for Diffraction-Limited Microscopy Data

Paper ID XXX

**Abstract.** Many microscopy applications are limited by the total amount of usable light and are consequently challenged by the resulting levels of noise in the acquired images. This problem is often addressed via (supervised) deep learning based denoising. Recently, by making assumptions about the noise statistics, self-supervised methods have emerged. Such methods are trained directly on the images that are to be denoised and do not require additional paired training data. While achieving remarkable results, self-supervised methods can produce high-frequency artifacts and achieve inferior results compared to supervised approaches. Here we present a novel way to improve the quality of self-supervised denoising. Considering that light microscopy images are usually diffraction-limited, we propose to include this knowledge in the denoising process. We assume the clean image to be the result of a convolution with a point spread function (PSF) and explicitly include this operation at the end of our neural network. As a consequence, we are able to eliminate high-frequency artifacts and achieve self-supervised results that are very close to the ones achieved with traditional supervised methods.

**Keywords:** denoising, CNN, light microscopy, deconvolution

## 1 Introduction

For most microscopy applications, finding the right exposure and light intensity to be used involves a trade-off between maximizing the signal to noise ratio and minimizing undesired effects such as phototoxicity. As a consequence, researchers often have to cope with considerable amounts of noise. To mitigate this issue, denoising plays an essential role in many data analysis pipelines, enabling otherwise impossible experiments [2].

Currently, deep learning based denoising, also known as content-aware image restoration (CARE) [24], achieves the highest quality results. CARE methods learn a mapping from noisy to clean images. Before being applied, they must be trained with pairs of corresponding noisy and clean training data.

In practice, this dependence on training pairs can be a bottleneck. While noisy images can usually be produced in abundance, recording their clean counterparts is difficult or impossible.

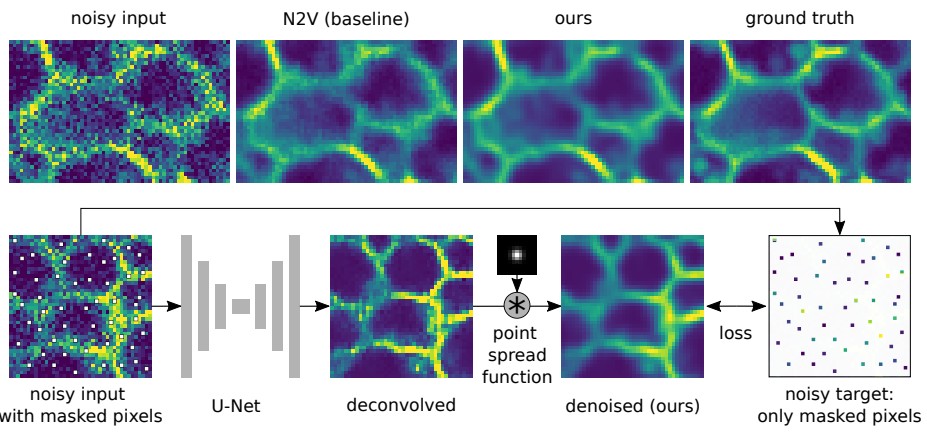

**Fig. 1. Improved Denoising for Diffraction-Limited Data. Top:** Given a noisy input, self-supervised methods like NOISE2VOID (N2V) [9] often produce high-frequency artifacts that do not occur in diffraction-limited data. Based on the assumption that the true signal must be the product of a convolution with a *point spread function* (PSF), our method is able to considerably improve denoising quality and remove these artifacts. **Bottom:** Our method is based on the NOISE2VOID masking scheme. Unpaired training images simultaneously serve as input and target. The loss is only calculated for a randomly selected set of pixels, which are masked in the input image. Our contribution is to convolve the output of the network with the PSF in order to produce a denoising result that is guaranteed to be consistent with diffraction-limited imaging. The output of the network before the convolution operation can be interpreted as a deconvolution result, which is a byproduct of our method. Our system can be trained in an end-to-end fashion, calculating the loss between our denoising result and the selected pixel set of the input image.

Over the last years, various solutions to the problem have been proposed. Lehtinen *et al.* showed that a network can be trained for denoising using only pairs of corresponding noisy images. This method is known as NOISE2NOISE [12].

The first self-supervised approaches NOISE2VOID [9] and NOISE2SELF [1] were introduced soon after this. These methods can be trained on unpaired noisy image data. In fact, they can be trained on the very same data that is to be denoised in the first place. The underlying approach relies on the assumption that (given the true signal) the noise in an image is generated independently for each pixel, as is indeed the case for the dominant sources of noise in light microscopy (Poisson shot noise and Gaussian readout noise) [13, 25]. Both methods employ so-called *blind spot* training, in which random pixels are masked in the input image with the network trying to predict their value from the surrounding patch.

Unfortunately, the original self-supervised methods typically produce visible high-frequency artifacts (see Figure 1) and can often not reach the quality achieved by supervised CARE training. It is worth noting that the high-frequency artifacts produced by these self-supervised methods never occur in

the real fluorescence signal. Since the image is diffraction-limited and oversampled, the true signal has to be smooth to some degree.

Multiple extensions of NOISE2VOID and NOISE2SELF have been proposed [10, 11, 17, 6]. All of them improve results by explicitly modeling the noise distribution.

Here, we propose an alternate and novel route to high-quality self-supervised denoising. Instead of making additional assumptions about the noise, we show that the result can be improved by including additional knowledge about the structure of our signal. We believe that our approach might ultimately complement existing methods that are based on noise modeling, to further improve denoising quality.

We assume that the true signal is the product of a convolution of an unknown *phantom image* and an approximately known point spread function (PSF) – a common assumption in established deconvolution approaches [20]. We use a U-NET [21] to predict the phantom image and then explicitly perform the convolution to produce the final denoised result (see Figure 1). We follow [9, 1] and use a blind spot masking scheme allowing us to train our network in an end-to-end fashion from unpaired noisy data.

We demonstrate that our method achieves denoising quality close to supervised methods on a variety of real and publicly available datasets. Our approach is generally on-par with modern noise model based methods [10, 16], while relying on a much simpler pipeline.

As a byproduct, our method outputs the predicted phantom image, which can be interpreted as a deconvolution result. While we focus on the denoising task in this paper, we find that we can produce visually convincing deconvolved images by including a positivity constraint for the deconvolved output.

## 2  Related work

In the following, we will discuss related work on self-supervised blind spot denoising and other unsupervised denoising methods. We will focus on deep learning-based methods and omit the more traditional approaches that directly operate on individual images without training. Finally, we will briefly discuss concurrent work that tries to jointly solve denoising and inverse problems such as deconvolution.

### 2.1  Self-Supervised Blind Spot Denoising

By now, there is a variety of different blind spot based methods. While the first self-supervised methods (NOISE2VOID and NOISE2SELF) use a masking scheme to implement blind spot training, Laine *et al.* [11] suggest an alternative approach. Instead of masking, the authors present a specific network architecture that directly implements the blind spot receptive field. Additionally, the authors proposed a way to improve denoising quality by including a simple pixel-wise Gaussian based noise model. In parallel, Krull *et al.* [10] introduced a similar

noise model based technique for improving denoising quality, this time using the pixel masking approach. Instead of Gaussians, Krull *et al.* use histogram-based noise models together with a sampling scheme. Follow-up work additionally introduces parametric noise models and demonstrates how they can be boot-strapped (estimated) directly from the raw data [17].

All mentioned methods improve denoising quality by modeling the imaging noise. We, In contrast, are the first to show how blind spot denoising can be improved by including additional knowledge of the signal itself, namely the fact that it is diffraction-limited and oversampled.

While the blind spot architecture introduced in [11] is computationally cheaper than the masking scheme from [9, 6], it is unfortunately incompatible with our setup (see Figure 1). Applying a convolution after a blind spot network would break the blind spot structure of the overall architecture. We thus stick with the original masking scheme, which is architecture-independent and can directly be applied for end-to-end training.

## 2.2  Other Unsupervised Denoising Approaches

An important alternative route is based on the theoretical work known as *Stein's unbiased risk estimator* (SURE) [22]. Given noisy observation, such as an image corrupted by additive Gaussian noise, Stein's 1981 theoretical work enables us to calculate the expected mean-squared error of an estimator that tries to predict the underlying signal without requiring access to the true signal. The approach was put to use for conventional (non-deep-learning-based) denoising in [18] and later applied to derive a loss function for neural networks [15]. While it has been shown that the same principle can theoretically be applied for other noise models beyond additive Gaussian noise [19], this has to our knowledge not yet been used to build a general unsupervised deep learning based denoiser.

In a very recent work called DivNoising [16] unsupervised denoising was achieved by training a variational autoencoder (VAE) [7] as a generative model of the data. Once trained, the VAE can produce samples from an approximate posterior of clean images given a noisy input, allowing the authors to provide multiple diverse solutions or to combine them to a single estimate.

Like the previously discussed extensions of blind spot denoising [11, 10, 17, 6] all methods based on SURE as well as DivNoising rely on a known noise model or on estimating an approximation. We, in contrast, do not model the noise distribution in any way (except assuming it is zero centered and applied at the pixel level) and achieve improved results.

A radically different path that does not rely on modeling the noise distribution was described by Ulyanov *et al.* [23]. This technique, known as *deep image prior*, trains a network using a fixed pattern of random inputs and the noisy image as a target. If trained until convergence, the network will simply produce the noisy image as output. However, by stopping the training early (at an adequate time) this setup can produce high-quality denoising results. Like our self-supervised method, deep image prior does not require additional training data to be applied. However, it is fundamentally different in that it is trained

and applied separately for each image that is to be denoised, while our method can, once it is trained, be readily applied to previously unseen data.

### 2.3  Concurrent Work on Denoising and Inverse Problems

Kobayashi *et al.* [8] developed a similar approach in parallel to ours. They provide a mathematical framework on how inverse problems such as deconvolution can be tackled using a blind spot approach. However, while we use a comparable setup, our perspective is quite different. Instead of deconvolution, we focus on the benefits for the denoising task and show that the quality of the results on real data can be dramatically improved.

   Yet another alternative approach was developed by Hendriksen *et al.* [5]. However, this technique is limited to well-conditioned inverse problems like computer tomography reconstruction and is not directly applicable to the type of microscopy data we consider here.

## 3  Methods

In the following, we first describe our model of the image formation process, which is the foundation of our method, and then formally describe the denoising task. Before finally describing our method for blind spot denoising with diffraction-limited data, we include a brief recap of the original NOISE2VOID method described in [9].

### 3.1  Image Formation

We think of the observed noisy image $\mathbf{x}$ recorded by the microscope, as being created in a two-stage process. Light originates from the excited fluorophores in the sample. We will refer to the unknown distribution of excited fluorophores as the *phantom image* and denote it as $\mathbf{z}$. The phantom image is mapped through the optics of the microscope to form a distorted image $\mathbf{s}$ on the detector, which we will refer to as *signal*. We assume the signal is the result of a convolution $\mathbf{s} = \mathbf{z} * \mathbf{h}$ between the phantom image $\mathbf{z}$ and a known PSF $\mathbf{h}$ [20].

   Finally, the signal is subject to different forms of imaging noise, resulting in the noisy observation $\mathbf{x}$. We think of $\mathbf{x}$ as being drawn from a distribution $\mathbf{x} \sim p_{\mathrm{NM}}(\mathbf{x}|\mathbf{s})$, which we call the *noise model*. Assuming that (given a signal $\mathbf{s}$) the noise is occurring independently for each pixel, we can factorize the noise model as

$$p_{\mathrm{NM}}(\mathbf{x}|\mathbf{s}) = \prod_{i}^{N} p_{\mathrm{NM}}(x_i, s_i), \qquad (1)$$

where $p_{\mathrm{NM}}(x_i, s_i)$ is the unknown probability distribution, describing how likely it is to measure the noisy value $x_i$ at pixel $i$ given an underlying signal $s_i$. Note that such a noise model that factorizes over pixels can describe the most dominant sources of noise in fluorescent microscopy, the Poisson shot noise and

readout noise [4, 25]. Here, the particular shape of the noise model does not have to be known. The only additional assumption we make (following the original NOISE2VOID [9]) is that the added noise is centered around zero, that is the expected value of the noisy observations at a pixel is equal to the signal $\mathbb{E}_{p_{\mathrm{NM}}(x_i,s_i)}[x_i] = s_i$.

### 3.2 Denoising Task

Given an observed noisy image $\mathbf{x}$, the denoising task as we consider it in this paper is to find a suitable estimate $\hat{\mathbf{s}} \approx \mathbf{s}$. Note that this is different from the deconvolution task, attempting to find an estimate $\hat{\mathbf{z}} \approx \mathbf{z}$ for the original phantom image.

### 3.3 Blind Spot Denoising Recap

In the originally proposed NOISE2VOID, the network is seen as implementing a function $\hat{s}_i = f(\mathbf{x}_i^{\mathrm{RF}}; \theta)$, that predicts an estimate for each pixel's signal $\hat{s}_i$ from its surrounding patch $\mathbf{x}_i^{\mathrm{RF}}$, which includes the noisy pixel values in a neighborhood around the pixel $i$ but excludes the value $x_i$ at the pixel itself. We use $\theta$ to denote the network parameters.

The authors of [9] refer to $\mathbf{x}_i^{\mathrm{RF}}$ as a *blind spot receptive field*. It allows us to train the network using unpaired noisy training images $x$, with the training loss computed as a sum over pixels comparing the predicted results directly to the corresponding values of the noisy observation

$$\sum_i \left(\hat{s}_i - x_i\right)^2. \tag{2}$$

Note that the blind spot receptive field is necessary for this construction, as a standard network, in which each pixel prediction is also based on the value at the pixel itself would simply learn the identity transformation when trained using the same image as input and as target.

To implement a network with a blind spot receptive field NOISE2VOID uses a standard U-NET [21] together with a masking scheme during training. The loss is only computed for a randomly selected subset of pixels $M$. These pixels are *masked* in the input image, replacing their value with a random pixel value from a local neighborhood. A network trained in this way acts as if it had a blind spot receptive field, enabling the network to denoise images once it has been trained on unpaired noisy observations.

### 3.4 Blind Spot Denoising for Diffraction-Limited Data

While the self-supervised NOISE2VOID method [9] can be readily applied to the data $\mathbf{x}$ with the goal of directly producing an estimate $\hat{\mathbf{s}} \approx \mathbf{s}$, this is a sub-optimal strategy in our setting.

Considering the above-described process of image formation, we know that, since **s** is the result of a convolution with a PSF, high-frequencies must be drastically reduced or completely removed. It is thus extremely unlikely that the true signal would include high-frequency features as they are *e.g.* visible in the NOISE2VOID result in Figure 1. While a network might in principle learn this from data, we find that blind spot methods usually fail at this and produce high-frequency artifacts.

To avoid this problem, we propose to add a convolution with the PSF after the U-NET (see Figure 1). When we now interpret the final output after the convolution as an estimate of the signal $\hat{\mathbf{s}} \approx \mathbf{s}$, we can be sure that this output is consistent with our model of image formation and can *e.g.* not contain unrealistic high-frequency artifacts.

In addition, we can view the direct output before the convolution as an estimate of the phantom image $\hat{\mathbf{z}} \approx \mathbf{z}$, *i.e.* an attempt at deconvolution.

To train our model using unpaired noisy data, we adhere to the same masking scheme and training loss (Eq. 2) as in NOISE2VOID. The only difference being that our signal is produced using the additional convolution, thus enforcing the adequate dampening of high-frequencies in the final denoising estimate.

### 3.5 A Positivity Constraint for the Deconvolved Image

Considering that the predicted deconvolved phantom image $\hat{\mathbf{z}}$ describes the distribution of excited fluorophores in our sample (see Section 3.1), we know that it cannot take negative values. After all, a negative fluorophore concentration can never occur in a physical sample.

We propose to enforce this constraint using an additional loss component, linearly punishing negative values. Together with the original NOISE2VOID loss our loss is computed as

$$\frac{1}{|M|} \sum_{i \in M} (\hat{s}_i - x_i)^2 + \lambda \frac{1}{N} \sum_{i=1}^{N} \max(0, -\hat{z}_i), \qquad (3)$$

where $N$ is the number of pixels and $\lambda$ is a hyperparameter controlling the influence of the positivity constraint. Note that the new positivity term can be evaluated at each pixel in the image, while the NOISE2VOID component can only be computed at the masked pixels.

## 4  Experiments and Results

In the following, we evaluate the denoising performance of our method comparing it to various baselines. Additionally, we investigate the effect of the positivity constraint (see Section 3.5). Finally, we describe an experiment on the role of the PSF used for reconstruction.

### 4.1   Datasets

**Fluorescence Microscopy Data with Real Noise.** We used 6 fluorescence microscopy datasets with real noise.

The *Convallaria* [10, 17] and *Mouse actin* [10, 17] datasets each consist of a set of 100 noisy images of $1024 \times 1024$ pixels showing a static sample. The *Mouse skull nuclei* [10, 17] consist of a set of 200 images of $512 \times 512$ pixels. In all 3 datasets, the ground truth is derived by averaging all images. We use all 5 images in each dataset for validation and the rest for training. The authors of [10, 17] define a region of each image that is to be used for testing, while the whole image can be used for training of self-supervised methods. We adhere to this procedure.

We additionally use data from [26], which provides 3 channels with training and test sets each consisting of 80 and 40, respectively. We use 15% of the training data for validation. Images are $512 \times 512$ pixels in size. Note that like [16] we use the raw data made available to us by the authors as the provided normalized data is not suitable for our purpose. The dataset provides 5 different versions of each image with different levels of noise. In this work, we use only the version with the minimum and maximum amount of noise. We will refer to them as *W2S avg1* and *W2S avg16* respectively, as they are created by averaging different numbers of raw images.

**Fluorescence Microscopy Data with Synthetic Noise.** Additionally, we use 2 fluorescence microscopy datasets from [3] and added synthetic noise. We will refer to them as *Mouse (Denoiseg)* and *Flywing (Denoiseg)*. While the original data contains almost no noise, we add pixel-wise Gaussian noise with standard deviation 20 and 70 for *Mouse (Denoiseg)* and *Flywing (Denoiseg)*, respectively. Both datasets are split into a training, validation, and test fraction. The *Mouse* dataset, provides 908 images of $128 \times 128$ pixels for training, 160 images of the same size as a validation set, and 67 images of $256 \times 256$ as a test set. The *Flywing* dataset, provides 1428 images size $128 \times 128$ as a training set, 252 images for validation (same size), and also 42 images size $512 \times 512$ as test set. As our method does not require ground truth, we follow [16] and add the test fraction to the training data in order to achieve a fair comparison.

**Synthetic Data.** While the above-mentioned datasets are highly realistic, we do not know the true PSF that produced the images. To investigate the effect of a mismatch between the true PSF and the PSF used in the training of our method, we used the clean rendered text data from the book *The beetle* [14] previously introduced in [16], synthetically convolved it using a Gaussian PSF with a standard deviation of 1 pixel width. Finally, we added pixel-wise Gaussian noise with a standard deviation of 100. The resulting data consists of 40800 small images of $128 \times 128$ pixels in size. We split off a validation fraction of 15%.

### 4.2   Implementation Details and Training

Our implementation is based on the *pytorch* NOISE2VOID implementation from [10]. We use the exact same network architecture, with the only difference being the added convolution with the PSF at the end of the network.

In all our experiments, we use the same network parameters: A 3-depth U-Net with 1 input channel and 64 channels in the first layer. All networks were trained for 200 epochs, with 10 steps per epoch. We set the initial learning rate to 0.001 and used Adam optimizer, batch size = 1, virtual batch size = 20, and patch size = 100. We mask 3.125% (the default) of pixels in each patch. We use the positivity constraint with $\lambda = 1$ (see Section 3.5).

### 4.3 Denoising Performance

We report the results for all fluorescence microscopy datasets in Table 1. The performance we can achieve in our denoising task is measured quantitatively by calculation of the average peak signal-to-noise ratio (**PSNR**). Qualitative results can be found in Figure 2.

We run our method using a Gaussian PSF with a standard deviation of 1 pixel width for all datasets. Figure 2 shows examples of denoising results on different datasets.

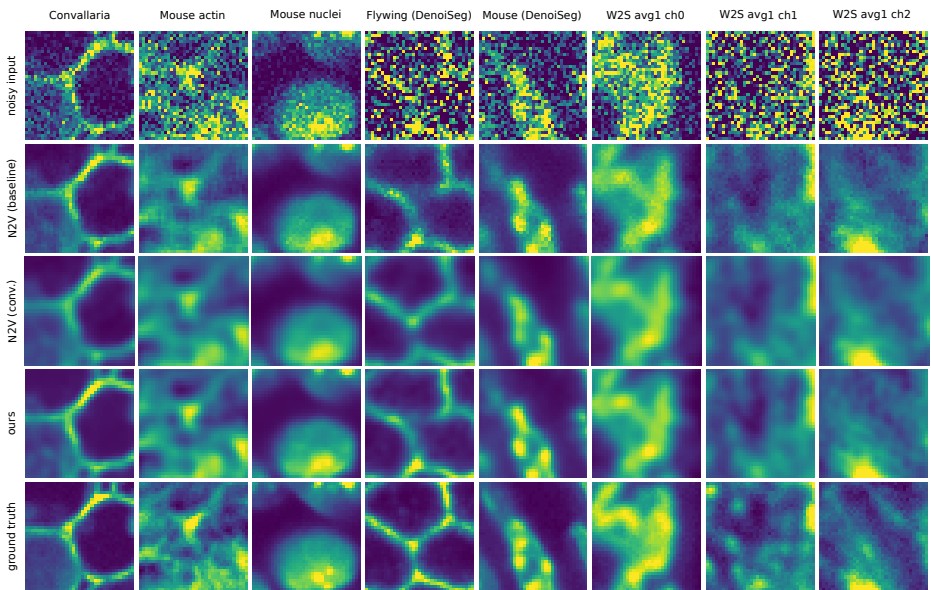

**Fig. 2. Denoising results.** We show cropped denoising results for various fluorescence microscopy datasets. Our method achieves considerable visual improvements for all datasets compared to Noise2Void. The *N2V (conv.)* baseline corresponds to the Noise2Void result convolved with the same PSF we use for our proposed method.

To assess the denoising quality of our method we compare its results to various baselines. We compared our method to Noise2Void, noise model based

| dataset/ network | raw data | self-supervised | | | | | | superv. |
| --- | --- | --- | --- | --- | --- | --- | --- | --- |
| | | no noise model | | | | noise model | | |
| | | N2V | N2V conv. | ours⁻, | ours⁺ | PN2V | DivN. | CARE |
| Convallaria | 28.98 | 35.85 | 32.86 | **36.39** | 36.26 | 36.47 | 36.94 | 36.71 |
| Mouse actin | 23.71 | 33.35 | 33.48 | 33.94 | **34.04** | 33.86 | 33.98 | 34.20 |
| Mouse nuclei | 28.10 | 35.86 | 34.59 | **36.34** | 36.27 | 36.35 | 36.31 | 36.58 |
| Flywing n0 | 11.15 | 23.62 | 23.51 | 24.10 | **24.30** | 24.85 | 25.10 | 25.60 |
| Mouse n0 | 20.84 | 33.61 | 32.27 | **33.91** | 33.83 | 34.19 | 34.03 | 34.63 |
| W2S avg1 ch0 | 21.86 | 34.30 | 34.38 | **34.90** | 34.24 | - | 34.13 | 34.30 |
| W2S avg1 ch1 | 19.35 | 31.80 | 32.23 | **32.31** | 32.24 | - | 32.28 | 32.11 |
| W2S avg1 ch2 | 20.43 | 34.65 | **35.19** | 35.03 | 35.09 | 32.48 | 35.18 | 34.73 |
| W2S avg16 ch0 | 33.20 | 38.80 | 38.73 | **39.17** | 37.84 | 39.19 | 39.62 | 41.94 |
| W2S avg16 ch1 | 31.24 | 37.81 | 37.49 | **38.33** | 38.19 | 38.24 | 38.37 | 39.09 |
| W2S avg16 ch2 | 32.35 | 40.19 | 40.32 | 40.60 | **40.74** | 40.49 | 40.52 | 40.88 |

**Table 1. Quantitative Denoising Results.** We report the average peak signal to noise ratio for each dataset and method. Here, *ours⁺* and *ours⁻* correspond to our method with ($\lambda = 1$) and without positivity constraint ($\lambda = 0$), see Section 3.5 for details. The best results among self-supervised methods without noise model are highlighted in bold. The best results overall are underlined. Here *DivN.* is short for DivNoising [16].

self-supervised methods (PN2V [10], DivNoising [16]), as well as the well-known supervised CARE [24] approach. While we run Noise2Void ourselves, the PSNR values for all other methods were taken from [16].

We created a simple additional baseline by convolving the Noise2Void result with the same PSF used in our own method. This baseline is referred to as *N2V (conv.)*.

## 4.4   Effect of the Positivity Constraint

Here we want to discuss the effect of the positivity constraint (see Section 3.5) on the denoising and deconvolution results. We compare our method without positivity constraint ($\lambda = 0$, see Eq. 3) and with positivity constraint ($\lambda = 1$). Choosing different values for $\lambda$ did not have a noticeable effect.

We find that the constraint does not provide a systematic advantage or disadvantage with respect to denoising quality (see Table 1). In Figure 3 we compare the results visually. While it is difficult to make out any differences in the denoising results, we see a stunning visual improvement for the deconvolution result when the positivity constraint is used. While the deconvolution result without positivity constraint contains various artifacts such as random repeating structures and grid patterns, these problems largely disappear when the positivity constraint is used.

We find it is an interesting observation that such different predicted phantom images can lead to virtually indistinguishable denoising results after convolu-

tion with the PSF, demonstrating how ill-posed the unsupervised deconvolution problem really is.

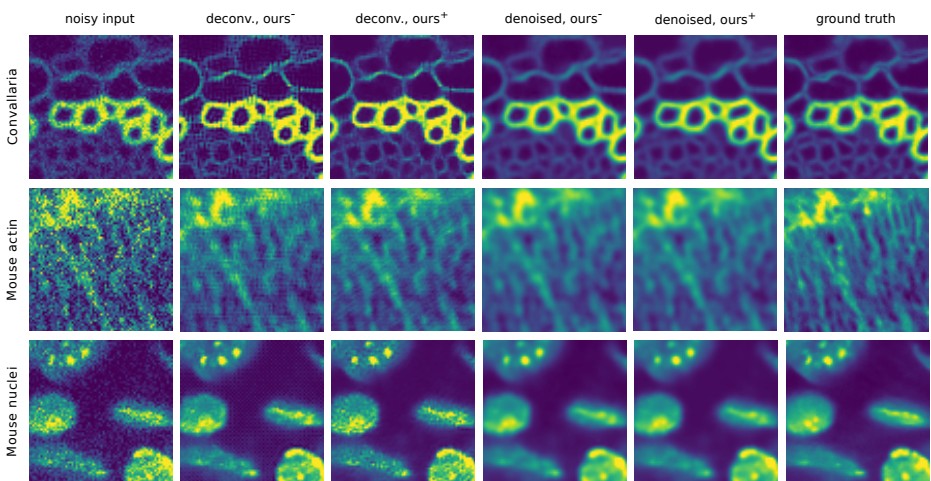

**Fig. 3. Effect of the proposed Positivity Constraint.** We show cropped denoising and deconvolution results from various datasets with (*ours+*) and without positivity constraint (*ours−*), see Section 3.5 for details. While the denoising results are almost indistinguishable, the deconvolution results show a drastic reduction of artifacts when the positivity constraint is used.

## 4.5    Effect of the Point Spread Function

Here we want to discuss an additional experiment on the role of the PSF used in the reconstruction and the effect of a mismatch with respect to the PSF that actually produced the data.

We use our synthetic *The beetle* dataset (see Section 4.1) that has been convolved with a Gaussian PSF with a standard deviation of $\sigma = 1$ pixel width and was subject to Gaussian noise of standard deviation 100. We train our method on this data using different Gaussian PSFs with standard deviations between $\sigma = 0$ and $\sigma = 2$. We used an active positivity constraint with $\lambda = 1$. The results of the experiment can be found in Figure 4.

We find that the true PSF of $\sigma = 1$ gives the best results. While lower values lead to increased artifacts, similar to those produced by NOISE2VOID, larger values lead to an overly smooth result.

# 5    Discussion and Outlook

Here, we have proposed a novel way of improving self-supervised denoising for microscopy, making use of the fact that images are typically diffraction-limited.

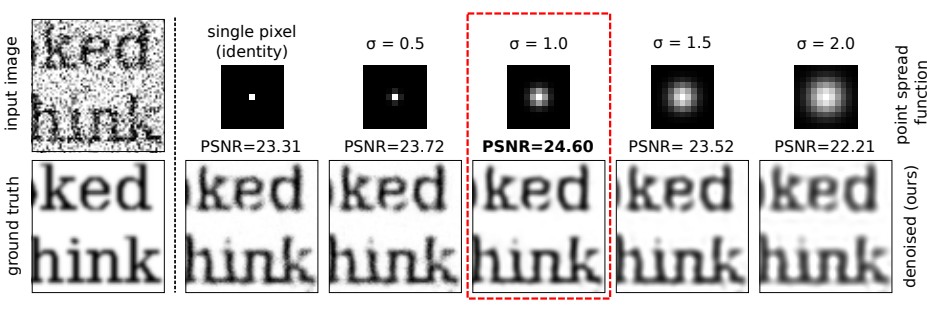

**Fig. 4. Effects of Point Spread Function Mismatch.** We use synthetic data to investigate how the choice of PSF influences the resulting denoising quality. The data was generated by convolving rendered text with a Gaussian PSF of standard deviation $\sigma = 1$ (highlighted in red) and subsequently adding noise. Here, we show the results of our method when trained using Gaussian PSFs of various sizes. We achieve the best results by using the true PSF. Smaller PSFs produce high-frequency artifacts. Larger PSFs produce overly smooth images.

While our method can be easily applied, results are often on-par with more sophisticated second-generation self-supervised methods [10, 16]. We believe that the simplicity and general applicability of our method will facilitate fast and widespread use in fluorescence microscopy where oversampled and diffraction-limited data is the default. While the standard deviation of the PSF is currently a parameter that has to be set by the user, we believe that future work can optimize it as a part of the training procedure. This would provide the user with an *de facto* parameter-free turn-key system that could readily be applied to unpaired noisy raw data and achieve results very close to supervised training.

In addition to providing a denoising result, our method outputs a decon-volved image as well. Even though deconvolution is not the focus of this work, we find that including a positivity constraint in our loss enables us to predict visually plausible results. However, the fact that dramatically different predicted deconvolved images give rise to virtually indistinguishable denoising results (see Figure 3) illustrates just how underconstrained the deconvolution task is. Hence, further regularization might be required to achieve deconvolution results of op-timal quality. In concurrent work, Kobayashi *et al*. [8] have generated decon-volution results in a similar fashion and achieved encouraging results in their evaluation. We expect that future work will quantify to what degree the positiv-ity constraint and other regularization terms can further improve self-supervised deconvolution methods.

We believe that the use of a convolution after the network output to account for diffraction-limited imaging will in the future be combined with noise model based techniques, such as the self-supervised [10, 11] or with novel techniques like DivNoising. In the latter case, this might even enable us to produce diverse deconvolution results and allow us to tackle uncertainty introduced by the under-constrained nature of the deconvolution problem in a systematic way.

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
