# OpenReview forum: "Improving Blind Spot Denoising for Diffraction-Limited Microscopy Data"
_thecvf.com/ECCV/2020/Workshop/BIC — BIC 2020 Oral_

### Official Review · AnonReviewer3 · 2020-07-29
**Theoretically and conceptually very strong, but I have concerns regarding the performance**

**Rating:** 6
**Confidence:** 4

**Review:**

In this work, the authors present an extension to the Noise2Void denoising framework that incorporates convolution with a point spread function in order to better approximate the image formation process in microscopy.

### Quality
The premise of the work is very solid and represents a shift towards making denoising approaches specific to bioimaging data rather than just the direct translation of computer vision techniques originally formulated for e.g. photographs, video data. This is an important conceptual advancement in the field.

The performance of the new method is assessed in comparison to a selection of other denoising frameworks, and when measured by the PSNR metric is shown to out-perform comparable self-supervised methods. However, I find the general reliance on the PSNR as a performance assessment to be problematic, as this does not take into account the structural content of the images post-denoising. For example, in Fig. 2 there appears to be some structural discrepancies in the ‘Flywing’ data. While the ‘N2V (conv.)’ image has a lower PSNR than ‘ours’, the visual agreement between the N2V (conv.) and ground truth data appears better than that between ours and ground truth. As a sanity check for myself, I thresholded and skeletonised these images and while the N2V (conv.) and ground truth skeletons matched well, the ‘ours’ skeleton deviated substantially at the central junction. Apologies if this seems facetious, but I think it underlines the necessity for another measure of performance, especially as the ultimate goal of denoising microscopy images is to produce a better baseline from which quantitative measurements of structure can be made (rather than just a visually pleasing image).
I would suggest that the authors remove the phrase ‘stunning visual improvement’ (line 443) as this is rather subjective – for example, using the positivity constraint in the mouse actin deconvolution does not improve the prevalence of patterned noise (which can be seen if the images are Fourier-transformed).

Section 4.5, wherein the effects of the PSF size are investigated, seems a little abrupt. Although the PSF size parameter is clearly critical for the performance of the method, this section would have benefitted from additional discussion of e.g. a non-uniform PSF throughout the image or tolerance to the PSF deviating from a Gaussian function, as these are both relevant considerations in real-life microscopy applications.

### Clarity
The paper is overall incredibly clear and I managed to understand the majority of what was written on my first pass (in contrast to my general experience reading papers in this field).

The repeated use of the phrase ‘diffraction-limited’ is somewhat misleading and may even be doing the work a disservice, This phrase is normally used in the context of referring to conventional widefield or confocal fluorescence imaging data; however there is no reason that the application of this method is limited to this regime. For example, given that the condition of a point spread function whose spatial distribution can be approximated by a Nyquist-sampled Gaussian distribution of known width, this approach could be readily applied to some super-resolution data such as STED images. For this reason, the authors may wish to reconsider the use of the phrase ‘diffraction-limited’ although this is just a suggestion.

### Originality
The work described here is a very similar concept to that described in [Kobayashi et al (2020)](https://arxiv.org/abs/2006.06156). However the authors acknowledge this work in their discussion and given that there was less than one month between the submission of the work by Kobayashi et al and the BIC submission deadline I do not see this as shortcoming in originality (rather, unfortunate timing). Setting aside the paper by Kobayashi et al, this paper displays interesting conceptual novelty. In comparison to the paper by Kobayashi et al, this work (in my opinion) is much better focused toward the application of fluorescence microscopy and is reported in such a way that I feel it is more likely that a microscopist would preferentially use the method presented here.

### Significance
The overall conceptual significance – integrating knowledge about the image formation process into the denoising method – is high, as I mentioned above. I am still not entirely convinced, however, that there is a *significant* increase in performance, as the PSNR values represent fairly marginal gains over to Noise2Void (alongside my above concerns regarding structural fidelity).

### Pros
* The paper is very well written, explained, and presented
* The theoretical benefits of the approach are substantial, and again the explanation of these is well-integrated into the paper
* The discussion of the paper shows that this work is a starting point and that the authors have thought about concrete ways to extend and improve it going forward.

### Cons
* The authors have not convinced me from a quantitative point of view that the results are superior to existing self-supervised methods.

**Reviews Visibility:**

I agree that my anonymized review is made publicly visible, if the submission is accepted.

---

### Official Review · AnonReviewer1 · 2020-07-31
**Interesting practical method but questionable concepts and results**

**Rating:** 6
**Confidence:** 4

**Review:**

The paper addresses the problem of denoising of microscopy images and the fact that traditional methods as well as various recent supervised deep learning methods make assumptions about the noise statistics that may not hold. The authors advocate the use of self-supervised deep learning methods, as high-quality paired training data is often not available to properly train supervised methods. But they observe that self-supervised methods typically produce high-frequency artifacts and achieve inferior results compared to supervised methods. To remedy this, they propose to exploit the fact that the images are usually diffraction-limited, by adding a convolution with a point-spread function model to an existing self-supervised deep learning-based denoising method (Noise2Void) and training it accordingly. Experimental results on a range of microscopy images illustrate the potential of the proposed method.

This paper is well written and the presentation is easy to follow. While the idea is interesting, I am not convinced it is theoretically sound. As explained (in Section 3.3 and also in Section 3.4), the Noise2Void method estimates the image s. Since s=z*h (Section 3.1), this makes it a denoising method, not a deconvolution method, and that is indeed how the method was designed. Thus, simply processing the estimated s by convolution with an assumed PSF model h (Figure 1 and Section 3.4) is questionable. Of course, doing so will force Noise2Void to behave more like it, and you can claim to "view the direct output before the convolution as an estimate of the phantom image ... i.e. an attempt at deconvolution" and get some visually pleasing results, but that does not make the approach theoretically right. Rather, it seems a practical trick that apparently happens to work to some extent.

Other specific comments:

- Section 4.1: Synthetic data is generated using a Gaussian PSF and pixel-wise additive Gaussian noise, but that is not realistic. As the authors admit elsewhere (multiple times), the dominant sources of noise are Poisson photon noise and Gaussian readout noise (Sections 1 and 3.1).

- Section 4.2: "Our implementation is based on the pytorch Noise2Void implementation from [10]. We use the exact same network architecture, with the only difference being the added convolution with the PSF at the end of the network." This, combined with the above major concern, make both the theoretical and the practical contribution of the paper rather limited.

- In Section 2 many methods are discussed but the comparison in Figure 2 is limited to only N2V (and a variant). Are there really no available software implementations of other methods to compare with?

- Section 4.3: The only quantitative measure used is PSNR. It is tricky to make the entire quantitative comparison hinge on a single measure that is known to be questionable. It would be good to also evaluate using other measures, such as SSIM.

- The authors claim "considerable visual improvements" (Figure 2) and even "stunning visual improvement (Section 4.4). These are subjective statements that in my opinion are not supported by the provided evidence.

**Reviews Visibility:**

I agree that my anonymized review is made publicly visible, if the submission is accepted.

---

### Decision · Program_Chairs · 2020-07-31

Accept (Oral)